# Selective whole-genome amplification reveals population genetics of *Leishmania braziliensis* directly from patient skin biopsies

Olivia A. Pilling[1], João L. Reis-Cunha[2], Cooper A. Grace[2], Alexander S. F. Berry[1], Matthew W. Mitchell[3], Jane A. Yu[4], Clara R. Malekshahi[1], Elise Krespan[1], Christina K. Go[1], Cláudia Lombana[1], Yun S. Song[4,5], Camila F. Amorim[1], Alexsandro S. Lago[6,7], Lucas P. Carvalho[6,7], Edgar M. Carvalho[6,7], Dustin Brisson[3], Phillip Scott[1], Daniel C. Jeffares[2], Daniel P. Beiting[1]*

**1** Department of Pathobiology, School of Veterinary Medicine, University of Pennsylvania, Philadelphia, Pennsylvania, United States of America, **2** Department of Biology, York Biomedical Research Institute, University of York, York, United Kingdom, **3** Department of Biology, School of Arts & Sciences, University of Pennsylvania, Philadelphia, Pennsylvania, United States of America, **4** Computer Science Division, University of California, Berkeley, Berkeley, California, United States of America, **5** Department of Statistics, University of California, Berkeley, Berkeley, California, United States of America, **6** Serviço de Imunologia, Complexo Hospitalar Universitário Professor Edgard Santos, Universidade Federal da Bahia, Salvador, Bahia, Brazil, **7** Laboratório de Pesquisas Clínicas do Instituto de Pesquisas Gonçalo Moniz, Fiocruz Bahia, Brazil

* beiting@upenn.edu

**Data Availability Statement:** Raw reads for all 18 SWGA genomes are available on the Sequence

## Abstract

In Brazil, *Leishmania braziliensis* is the main causative agent of the neglected tropical disease, cutaneous leishmaniasis (CL). CL presents on a spectrum of disease severity with a high rate of treatment failure. Yet the parasite factors that contribute to disease presentation and treatment outcome are not well understood, in part because successfully isolating and culturing parasites from patient lesions remains a major technical challenge. Here we describe the development of selective whole genome amplification (SWGA) for *Leishmania* and show that this method enables culture-independent analysis of parasite genomes obtained directly from primary patient skin samples, allowing us to circumvent artifacts associated with adaptation to culture. We show that SWGA can be applied to multiple *Leishmania* species residing in different host species, suggesting that this method is broadly useful in both experimental infection models and clinical studies. SWGA carried out directly on skin biopsies collected from patients in Corte de Pedra, Bahia, Brazil, showed extensive genomic diversity. Finally, as a proof-of-concept, we demonstrated that SWGA data can be integrated with published whole genome data from cultured parasite isolates to identify variants unique to specific geographic regions in Brazil where treatment failure rates are known to be high. SWGA provides a relatively simple method to generate *Leishmania* genomes directly from patient samples, unlocking the potential to link parasite genetics with host clinical phenotypes.

Read Archive (SRA) under accession number PRJNA875085. All code used for analysis of depth and breadth of coverage in SWGA samples, and annotation, analysis, and visualization of variants is available as a fully reproducible dockerized code "capsule" archived on Code Ocean (https://doi.org/10.24433/CO.3705597.v2).

**Funding:** This study was funded in part by grants from the National Institute of Allergy and Infectious Diseases (5R01AI143790 and 5R01AI149456-03 to PS, and 5T32AI007532-24 to OAP). The funders had no role in study design, data collection and analysis, decision to publish, or preparation of the manuscript.

**Competing interests:** The authors declare no competing interests.

## Author summary

*Leishmania braziliensis* is the main cause of cutaneous leishmaniasis in Brazil. Due to limitations in culturing, it is important to study the parasite in a culture-independent manner. We use selective whole genome amplification (SWGA) to explore parasite genomic diversity directly from patient biopsies. This method is inexpensive and can be broadly used to generate parasite genome sequence data from different *Leishmania* species infecting different mammalian hosts. We found high diversity among the *L. braziliensis* genomes from Bahia, Brazil, which correlated with geographic location. By integrating these data with publicly available genome sequences from other studies spanning four countries in South America, we identified variants unique to Northeast Brazil that may be linked to high regional rates of treatment failure.

## Introduction

*Leishmania* constitutes a genus of intracellular protozoan parasites whose species are all transmitted by the bite of an infected phlebotomine sand fly and can lead to leishmaniasis. This neglected tropical disease has a spectrum of clinical presentations, including visceral and cutaneous, which vary in severity and are influenced by parasite species and strain genetics [1,2]. The most common form of disease caused by these parasites is cutaneous leishmaniasis (CL), which is characterized by one or more localized skin ulcers. Moreover, up to 10% of patients can develop more severe forms of the disease, such as mucosal (ML) or disseminated leishmaniasis (DL) [2]. Worldwide there are 700,000 to 1 million new cases of CL annually [3]. Although mortality is low for patients with CL, the disease is disfiguring, leads to chronic and systemic inflammation [4], and adversely impacts quality of life.

In Brazil, CL cases are largely caused by *Leishmania braziliensis*. Previous population genetics studies of this species have relied on low-resolution techniques, such as multilocus sequence typing and restriction fragment length polymorphism, both of which only consider a small set of genetic loci. Collectively, these studies have shown that the genetic diversity of *L. braziliensis* is higher in and around the Amazon rainforest than near the coast [5,6]. Moreover, recent whole genome sequencing studies have determined that *L. braziliensis* exhibits higher intraspecies genetic variation than other *Leishmania* species [7,8]. Variation in virulence, drug resistance, and clinical phenotype among strains has been observed in many parasites. A recent study using random amplified polymorphic DNA analysis showed that *L. braziliensis* genotypes are associated with disease presentation in patients [9]. Collectively, these studies underscore the importance of generating high-resolution genotyping data from *L. braziliensis* to identify genetic variants linked to disease severity and treatment outcome in CL patients.

We recently showed that *L. braziliensis* burden in patients is a strong predictor of inflammation, pathology, and poor response to chemotherapy, yet the parasite factors that contribute to differences in parasite load between patients have been difficult to address [10]. Technical and biological factors associated with culture adaptation of *L. braziliensis* and limited economic resources in endemic regions further complicate efforts to generate high-resolution genomic data from this important species. Unlike other *Leishmania* species, *L. braziliensis* is characterized by relatively slow growth and low parasitemia, which pose a major challenge to isolating parasites from patient lesions [11,12]. Even when parasites are successfully adapted to culture, some studies suggest that drug resistance markers identified from *in vitro* assays may not be driving drug resistance observed in the clinic, and the process of isolating parasites from primary patient samples may transiently alter chromosomal copy number [13–15].

There is an urgent need for culture-independent methods to circumvent these issues. Since parasitemia at the site of infection in the skin is extremely low during *L. braziliensis* infection, a direct metagenomic sequencing approach is not a viable alternative to culture. Enrichment of *Leishmania donovani* genomes from primary patient samples was recently published using Agilent SureSelect arrays which utilize custom RNA 'bait' sequences to capture *Leishmania* genomic DNA for subsequent amplification [15]. However, this method is expensive, requires specialized reagents, and is specific to parasites causing visceral leishmaniasis (*L. donovani* and *L. infantum*).

In this study, we develop a selective whole genome amplification (SWGA) protocol to selectively amplify *L. braziliensis* directly from primary patient samples. SWGA is based on the use of organism-specific, short oligonucleotide primers and a high-fidelity, highly processive polymerase to preferentially amplify large segments of the target genome. Effective SWGA protocols have resulted in sequencing-ready samples that are enriched for specific target microbial genomes and which have been used to address biologically important questions in several microorganisms, including *Mycobacterium tuberculosis*, *Wolbachia* spp., *Plasmodium* spp., *Neisseria meningitidis*, *Coxiella burnetii*, *Wuchereria bancrofti*, and *Treponema pallidum* [16–30]. The ability to carry out SWGA without specialized equipment or reagents makes it feasible to implement in low- and middle-income countries (LMICs) where laboratory resources may be limited [16,21]. Here we report the development of SWGA for *Leishmania* and show that this method enables robust amplification of *L. braziliensis* DNA from complex metagenomic samples obtained from patients and experimental mouse models of infection. We investigate SNPs, indels, and somy in the parasite genomes directly sequenced from primary patient samples. Ultimately, we reveal the population genetic structure of *L. braziliensis* in Corte de Pedra, Bahia, Brazil, and compare these genomes to previously published *L. braziliensis* genomes from across S. America.

## Results

### Validation of SWGA for *Leishmania in silico* and using synthetic controls

We used the improved SWGA algorithm, swga2.0, which employs machine learning to design primer sets that preferentially bind to a target genome, compared to one or more background genomes ([31]; see Methods). We used *L. braziliensis* (MHOM/BR/75/M2904 2019) as the target genome and the human genome as background. Genomes from *Staphylococcus aureus* and *Streptococcus pyogenes* were also included as background since both are skin commensals that we previously reported to be common members of the dysbiotic skin microbiome on *L. braziliensis* lesions [32]. We calculated the expected number of perfect match binding sites–across a range of parasite and host genomes–for each of the 23, 8-mer primer sequences designed by the SWGA algorithm (S1 Table). This *in silico* analysis showed that our SWGA primers had a median of 15 (8.4–27.5) 'hits', or exact matches, per million base pairs (Mbp) of the *L. braziliensis* genome and a median of 0.22 (0.16–0.60) hits per Mbp of the human genome (Fig 1A)–a nearly 60-fold (27- to 100-fold) enrichment in predicted binding to the parasite genome compared to host (Fig 1B). We next tested whether our SWGA primers would be predicted to work when applied to other *Leishmania* species and/or when other host species were involved. Multiple species of *Leishmania* cause disease in humans, and several infect canines that are sympatric with humans. In addition, many *Leishmania* species are used to experimentally infect rodent models for research. *L. major*, *L. donovani*, *L. infantum*, and *L. amazonensis* all exhibited similar results with our SWGA 8-mers as *L. braziliensis*, with median hits per Mbp of 16.8, 15.5, 16.0, and 16.3, respectively (Fig 1A). Similarly, when our primers were tested against mouse or canine reference genomes, we observed 53-fold and 34-fold enrichment,

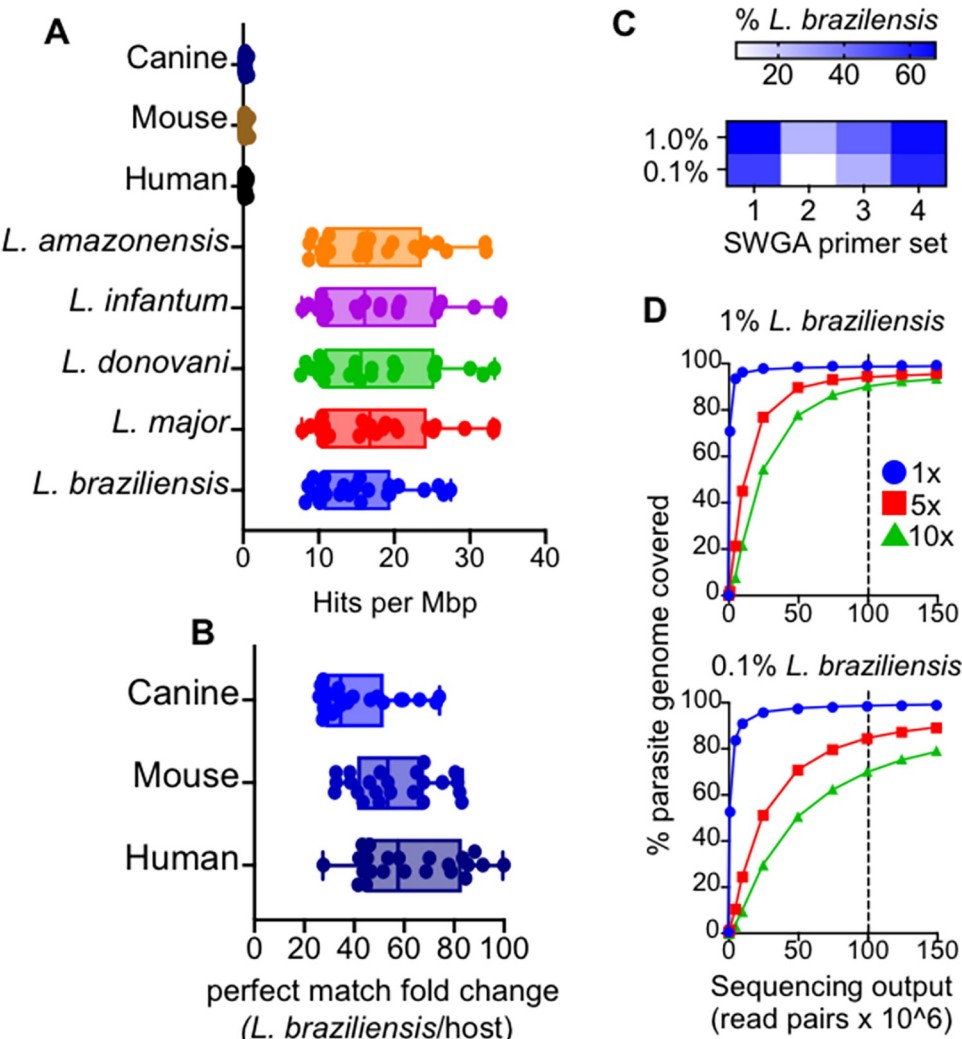

**Fig 1. SWGA primer design and evaluation.** (A) The number of exact match 'hits' per megabase (Mbp) for each of the 23 identified SWGA primers against *Leishmania* and host reference genomes, and (B) the fold difference in exact matches against *L. braziliensis* compared to human, mouse, or canine genomes. (C) Heatmap showing percent reads aligning to *L. braziliensis* for each of the four SWGA primer sets used to carry out SWGA on known ratios of *L. braziliensis* DNA spiked into human genomic DNA (0.1 and 1% final parasite DNA). (D) The number of reads is shown in relation to the percentage of the parasite genome covered at ≥1x (blue line), 5x (red line) and 10x (green line). Vertical dashed line indicates a sequencing effort of 100 x 10^6 150bp paired-end reads.

respectively, of predicted primer binding to the *L. braziliensis* genome over these hosts (**Fig 1B**). Taken together, these *in silico* data suggest that primers designed using SWGA are valuable in a wide range of contexts, from natural infection of humans and canines to experimental infections of mice.

 *L. braziliensis* is known to be present at low levels in skin lesions. Less than 1% of total reads from RNA-seq studies of lesions map to the parasite [10]. To evaluate the efficacy of our SWGA assay in a controlled setting that mimics patient samples, we prepared purified human DNA spiked with either 1% or 0.1% (w/w) purified *L. braziliensis* genomic DNA. Using high-throughput sequencing, we evaluated the ability of four separate SWGA primer sets, each consisting of 10 SWGA primers, to selectively amplify parasite DNA in these synthetic samples. After a 16-hour isothermal SWGA reaction, we found that multiple primer sets resulted in

substantial amplification of the synthetic samples. Primer set 1 (PS1) and PS4 yielded the best results, achieving ≥60% of parasite-mapping reads in samples that started with only 1% or 0.1% *L. braziliensis* DNA (**Fig 1C**). We next examined the depth and breadth of coverage following SWGA of these synthetic samples. For each sample, sequencing data from individual SWGA reactions (PS1, PS2, PS3, PS4) were combined and mapped to the parasite genome, and depth and breadth of coverage were evaluated at different sequencing efforts. In the 1% spike-in control, after SWGA, a sequencing effort of ~100M paired-end reads (**Fig 1D, top, vertical dashed line**) yielded 10x coverage across nearly 90% of the parasite genome, and 5x coverage across over 94% of the genome. Similarly, when the synthetic sample containing only 0.1% parasite DNA was used, the same sequencing effort resulted in 10x coverage across over 70% of the parasite genome and 5x coverage across 84% of the genome (**Fig 1D, bottom, vertical dashed line**). These data show that even when *L. braziliensis* DNA is present at incredibly low levels, and in the presence of abundant contaminating human DNA, SWGA yields an excellent breadth of coverage across the 32Mbp parasite genome.

## Validation of SWGA assay on mouse and primary human samples

Based on our *in silico* analysis (**Fig 1A and 1B**), we predicted that our SWGA primer sets would be effective in other species of *Leishmania*, as well as in other host backgrounds. To formally test this, we infected mice with either *L. braziliensis* (same target parasite species, but different host species background) or *L. major* (different parasite and different background) and carried out SWGA on DNA extracted from whole ears recovered from these mice. Tissues from experimentally infected mice have nearly undetectable levels of parasite sequences prior to SWGA (**Fig 2A**). After SWGA, however, the proportion of parasite reads increased to over 20% in one animal infected with *L. braziliensis* and three animals infected with *L. major* (**Fig 2A, circles and triangles, respectively**). Parasite burdens with *L. major* are generally higher than with *L. braziliensi*s, suggesting that SWGA is more effective as parasite burden increases.

We next tested our SWGA protocol on primary patient samples. DNA extracted from skin punch biopsies from 16 *L. braziliensis* patients was subjected to high-throughput sequencing before and after SWGA. Reads from these pre- and post-SWGA samples for each patient were mapped to the parasite genome to evaluate depth and breadth of coverage. Direct sequencing of DNA extracted from lesions showed that less than 0.5% of reads mapped to the parasite before SWGA (**Fig 2B, 'pre'**), consistent with the mouse data above and previous reports of extremely low parasite burden in *L. braziliensis* lesions [10]. However, following SWGA, these same samples showed dramatic increases in the proportion of parasite-mapping reads, ranging from 2% to 55%, with over half of the patient samples (9/16) having ≥ 20% of reads mapping to the parasite (**Fig 2B, 'post', dashed line**). To better evaluate the specificity of our SWGA primer design in the context of primary patient samples, we measured read mapping in pre- and post-SWGA from these 9 samples. Specifically, we evaluated reads mapping to the foreground genome (*L. braziliensis* nuclear), background genome (human, *Staphylococcus aureus*, and *Streptococcus pyogenes*), and genome sequences excluded from SWGA primer design (*L. braziliensis* kinetoplast genome). Prior to SWGA, *S. aureus*, *S. pyogenes*, and *Leishmania* kineotplast maxicircle combined accounted for less than 0.005% of the reads (**S1 Fig**). Following SGWA, the proportion of human reads decreased to around 55%, *L. braziliensis* reads increased to around 45%, while reads mapping to bacterial and *Leishmania* maxicircle reads remain below 0.1% (**S1 Fig**). These data show that SWGA is highly specific for the nuclear genome of *Leishmania*.

Next, we selected SWGA data from a single patient sample (**#7; Fig 2B, blue point**) and measured coverage across the parasite genome (**Fig 2C**), which showed that over 80% of the

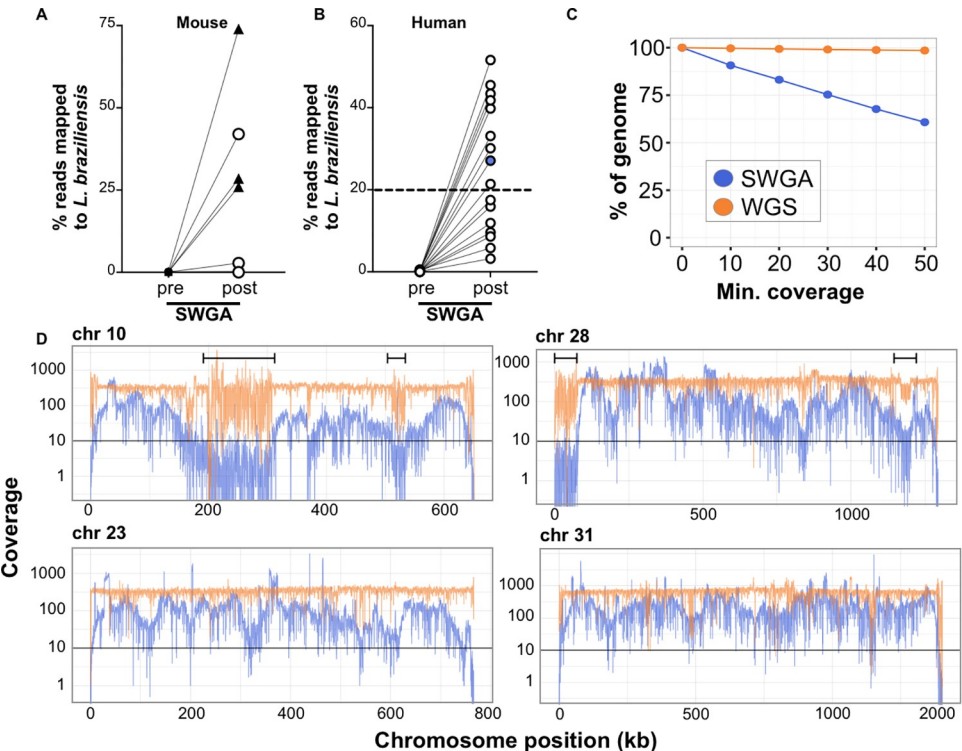

**Fig 2. In vivo validation of SWGA for *Leishmania*.** Percentage of reads mapping to *L. braziliensis* genome in DNA from (A) infected mouse ears (n = 3 animals infected with *L. major*, triangles; n = 5 animals infected with *L. braziliensis*, circles) or (B) patient lesion biopsies, sequenced before (pre) and after (post) SWGA. Data shown are from the SWGA primer set the yielded the best amplification for each sample. (C) Genome coverage for SWGA data from a single patient sample (patient #7, blue point from panel B). (D) Coverage of four selected *L. braziliensis* chromosomes in SWGA data from a single patient (#7; blue lines) compared to whole genome sequencing (WGS) of pure, cultured *L. braziliensis* (orange lines). Data shown in panel C and D are merged from all SWGA primer sets to maximize coverage.

genome was covered at 10x depth by SWGA, and over 50% of the genome at 50x. Based on these data, we reasoned that SWGA may be a useful tool for monitoring parasite genotypes linked to drug resistance and disease phenotypes. SWGA data from the same patient sample was evaluated for coverage across the full length of each of the 35 parasite chromosomes (S2 Fig). We focused our initial analysis on chromosomes 10, 23, and 31 since they encode the GP63, MRPA, and AQP1 genes, respectively, which have previously been linked to drug resistance in other *Leishmania* species [33–36]. In addition, chromosome 31 is known to have extra copies that have been linked to parasite adaptation to stress [37–39]. Lastly, we examined chromosome 28 because it has been linked to atypical manifestations of CL [40,41]. We observed over 10x coverage across most of the length of each of these chromosomes following SWGA (**Fig 2D, blue lines**). Regions that showed the poorest coverage in our SWGA samples often corresponded to ends of chromosomes or to regions (**Fig 2D, brackets**) that were also poorly covered in whole genome sequencing (WGS) of pure cultures of *L. braziliensis* (**Fig 2D, orange lines**). This result likely reflects low complexity regions that pose a challenge to genome sequencing for *L. braziliensis*, rather than issues specific to SWGA. Fluctuations in coverage observed in our SWGA data across the chromosome may be due to SWGA-inherent multiple displacement amplification (MDA), which makes it impossible to parse apart quantitative genetic information like gene copy number variation. These data indicate that SWGA directly applied to primary patient samples generates high-quality data suitable for high-resolution parasite genotyping.

## Somy analysis with SWGA

*Leishmania* parasites exhibit mosaic aneuploidy, and it has been suggested that modulating chromosomal copy number provides the parasite with a mechanism for regulating gene dosage in the absence of promoter-driven gene expression [42,43]. Previous attempts to use allele frequency to estimate somy of *L. infantum* were unsuccessful due to a low number of heterozygous SNPs in this parasite species [44]. Since *L. braziliensis* has been reported to have a higher number of SNPs than other *Leishmania* species [7], we tested whether the alternate allele read depth proportion (AARDP), as determined by SWGA, could be used to infer chromosome copy number. We first examined AARDP in DNA isolated from pure *L. braziliensis* cultures and subjected to either traditional WGS or SWGA (**Fig 3A and 3B, respectively**). Allele read depth distributions can be influenced by differential variations in chromosome copies within the cell population from a sample. Nevertheless, we still observed sharp peaks in the WGS sample centered over an AARDP of 0.5, suggesting that chromosomes 10, 23, and 28 were disomic (**Fig 3A**). In contrast, three distinct peaks were observed for chromosome 31 centered on an allele frequency of 0.25, 0.50, and 0.75, consistent with multiple previous reports that this chromosome is supernumerary, and potentially tetrasomic [37]. SWGA of the same pure culture closely resembled the WGS data, albeit with allele frequency peaks that were slightly less sharp (**Fig 3B**). SWGA of synthetic spike-in controls showed a similar profile at 1% parasite DNA (**Fig 3C**), which was diminished when parasite DNA dropped to 0.1% (**Fig 3D**), suggesting that extremely low parasite abundance will adversely impact the utility of SWGA for somy estimation. AARDP analysis of SWGA data from two patient samples showed evidence of a supernumerary state for chromosomes 23 in patient 7 (potentially tetrasomic; **Fig 3E**) and chromosome 28 in patient 61 (potentially trisomic; **Fig 3F**).

## A high-throughput screen of patient samples using SWGA

Routine diagnosis of *L. braziliensis* infection is carried out by collection of a punch biopsy from the site of the skin lesion followed by DNA extraction and parasite-specific PCR. We reasoned that this original DNA extract from a diagnostic biopsy, which is often archived for retesting purposes, could be sufficient for large-scale generation of parasite genomes by SWGA. To test this, we devised a screening approach that allowed us to scale our SWGA assay by an order of magnitude. 165 archived patient samples, of which 51 were intact skin biopsies and 114 were diagnostic DNA samples, were acquired from the health clinic in Corte de Pedra, Brazil. We anticipated that successful SWGA reactions would be positively correlated with parasite burden, therefore, our screen involves first prioritizing samples for SWGA using a parasite-specific qPCR [45] (**Fig 4A**). Based on qPCR results, 66 patient samples with the highest parasite burden were selected for SWGA (**S2 Table**). SWGA reactions were then arrayed in 96-well plates using different SWGA primer sets (**Fig 4B**). Since PS2 and PS3 performed more poorly on synthetic samples (**Fig 1C**), we chose to use these primer sets only in second-round SWGA reactions that had first undergone an initial round of SWGA with PS1 or PS4. These 'nested' SWGA reactions aim to amplify greater breadth of the parasite genome. Following SWGA, sequencing libraries were prepared, pooled, and subjected to shallow sequencing (**Fig 4C**). For each patient sample, all SWGA reactions yielding ≥20% reads mapping to the parasite from a shallow sequencing run were considered successful. The corresponding libraries were re-pooled (**Fig 4D**) and subjected to re-sequencing (**Fig 4E**). This screen of 66 patient samples yielded parasite genomes from 18 patients (27% success rate) with a median percentage of the parasite genome covered at ≥10x of 86.6% and a median genome coverage of 38x (**Fig 4F**). Given broad coverage across each of the 18 parasite genomes generated by SWGA, we next sought to call single nucleotide polymorphisms (SNPs) and

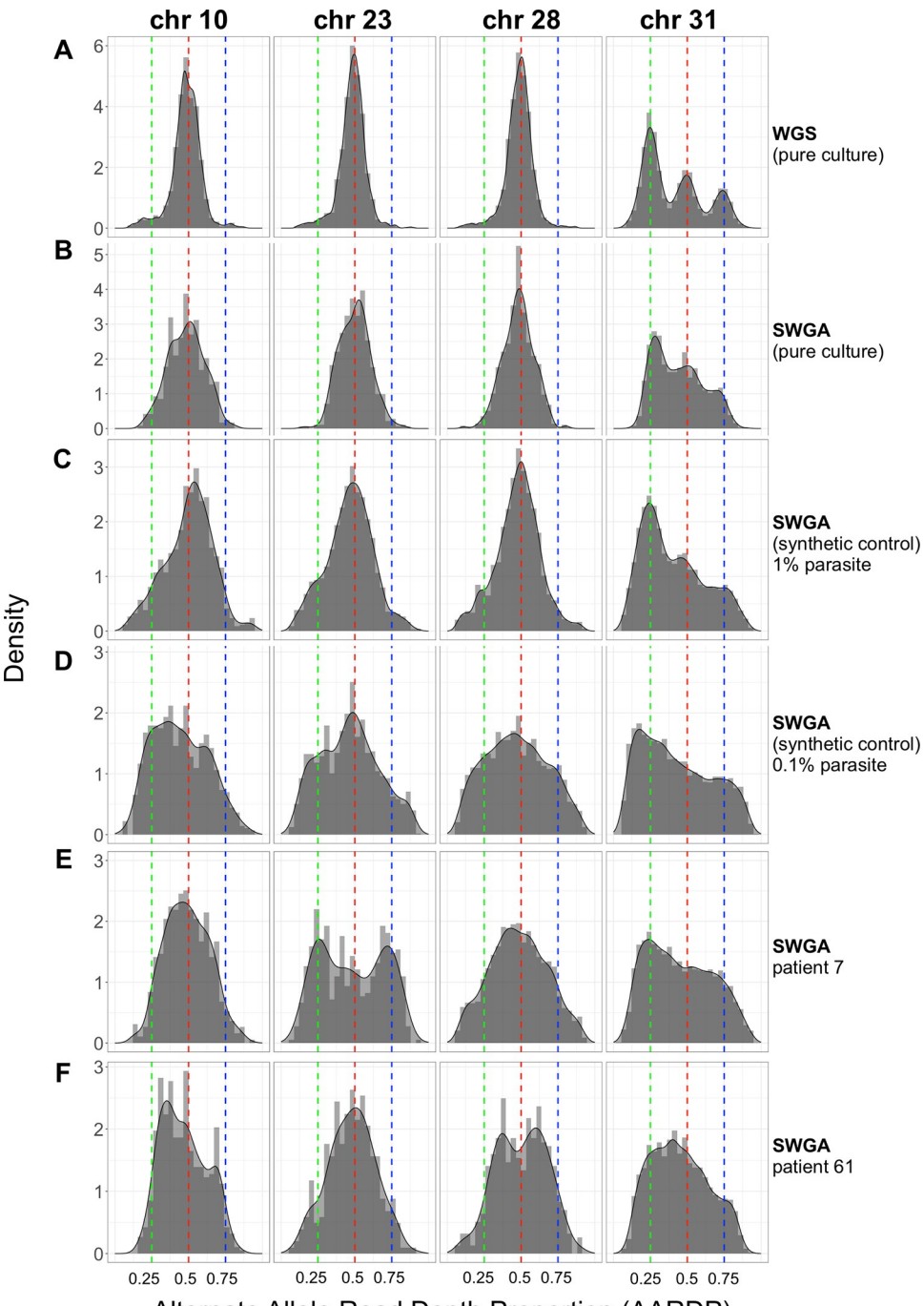

**Fig 3. Allele frequency determined by SWGA.** Alternate Allele Read Depth Proportion (AARDP) histograms for *L. braziliensis* chromosomes 10, 23, 28, and 31, for (A) whole genome sequencing (WGS) of pure cultured parasites, (B) SWGA of pure cultures, (C-D) SWGA of synthetic controls consisting of 1% (C) or 0.1% (D) parasite DNA, (E-F) SWGA on two patient samples from Fig 2B. Peaks centered on 0.5 indicate disomic chromosomes, while peaks at approximately 0.25, 0.5 and 0.75 indicate tetrasomic chromosomes. Green, red, and blue dashed lines denote an AARDP of 0.25, 0.5, and 0.75, respectively.

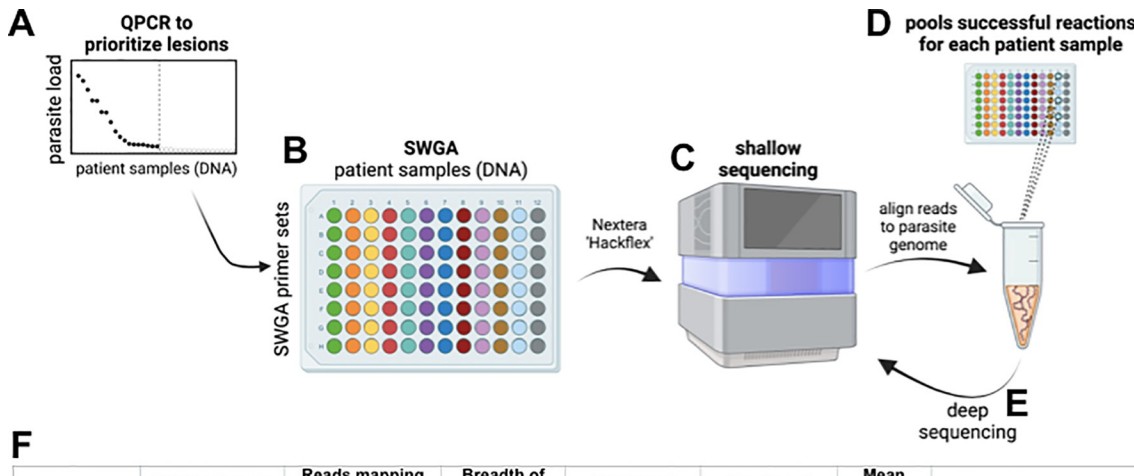

| Patient ID | Total # of reads (millions) | Reads mapping to *L. braziliensis* (millions) | Breadth of coverage at 10x (%) | Number of SNPs | Number of INDELs | Mean genome coverage | Municipality |
|---|---|---|---|---|---|---|---|
| 7 | 121.79 | 52.67 | 89.8 | 91947 | 18063 | 48 | Teolandia |
| 8 | 143.73 | 52.26 | 89.8 | 86158 | 17742 | 58 | Teolandia |
| 9 | 96.58 | 46.66 | 78.9 | 76775 | 15884 | 27 | Teolandia |
| 11 | 112.83 | 65.41 | 86.8 | 82875 | 17852 | 36 | Wenceslau Guimarães |
| 13 | 87.06 | 49.94 | 82.6 | 81576 | 16761 | 34 | Presidente Tancredo Neves |
| 15 | 101.86 | 49.47 | 84.4 | 80523 | 17418 | 38 | Presidente Tancredo Neves |
| 17 | 198.61 | 29.36 | 89.1 | 89145 | 19842 | 31 | Nilo Peçanha |
| 21 | 92.05 | 32.46 | 87.5 | 90188 | 18446 | 39 | Valença |
| 24 | 176.78 | 40.84 | 88.1 | 84495 | 20746 | 38 | Wenceslau Guimarães |
| 31 | 66.34 | 45.50 | 88.7 | 86391 | 17832 | 42 | Teolandia |
| 36 | 47.91 | 25.80 | 82.4 | 83444 | 17093 | 32 | Nilo Peçanha |
| 39 | 143.65 | 69.89 | 80.8 | 78018 | 18244 | 27 | Wenceslau Guimarães |
| 40 | 151.37 | 57.36 | 77.3 | 73011 | 16134 | 26 | Teolandia |
| 56 | 122.04 | 49.61 | 87.6 | 85569 | 20017 | 42 | Ubaira |
| 58 | 93.00 | 49.44 | 86.0 | 85759 | 17862 | 39 | Valença |
| 61 | 95.96 | 24.31 | 86.7 | 91811 | 18743 | 38 | Wenceslau Guimarães |
| 64 | 95.10 | 42.91 | 86.5 | 90288 | 17969 | 38 | Taperoá |
| 66 | 123.36 | 66.22 | 69.8 | 74259 | 14773 | 18 | Wenceslau Guimarães |
| Median (± SD) | 107.35 (± 37.48) | 49.46 (± 13.22) | 86.60 (± 5.22) | 85032 (± 5779) | 17857 (± 1468) | 38 (± 9) | |

**Fig 4. Scalable SWGA profiling of patient samples.** (A) QPCR is used to prioritize samples that have the highest parasite burden and, therefore, the greatest likelihood of success for SWGA. (B) SWGA is carried out in 96-well plates using multiple primer sets and primer set combinations (plate rows) for each patient (plate columns). (C) Shallow sequencing is used to determine which samples showed the best amplification by SWGA. (D) All successful SWGA reactions are pooled for each patient and (E) subjected to deep sequencing. (F) Results of selective whole genome amplification of *L. braziliensis* from 18 primary patient samples.

insertions/deletions (INDELs) against the reference *L. braziliensis* genome. Across all 18 SWGA-generated genomes we observed a median of 85,032 SNPs and 17,857 INDELs (**Fig 4F**), a finding that is consistent with the number of SNPs/INDELs previously reported in genome sequences from cultured isolates of *L. braziliensis* [7].

## Integrating SWGA and WGS genomes for population genomics of *L. braziliensis* in South America

Several *L. braziliensis* genomes have been generated from cultured parasite isolates, which prompted us to ask whether SWGA generates genomes of sufficient quality to compare with isolate data for large-scale population genomic studies. We carried out an integrated analysis of our 18 *L. braziliensis* SWGA genomes together with 41 publicly available *L. braziliensis* genomes generated from cultured isolates, including 4 from Bahia, Brazil [46], 10 from

Pernambuco, Brazil [7], 1 from Rondônia, Brazil [47], 18 from Peru [47], 6 from Colombia [8], and 2 from Bolivia [8,47]. Collectively, these 59 genomes span a wide geographic range (**Fig 5A**), with our SWGA samples contributing genomes from areas of Bahia, Brazil that were not previously covered by other studies (**Fig 5B**). Principal component analysis (PCA) of SNP data from these genomes shows clear separation by geographic location (**Fig 5C**), with *L. braziliensis* genomes from Brazil clustering tightly together (**Fig 5C, upper right**) but distinct from Colombian, Peruvian, and Bolivian isolates. Two genomes from a forested region of Brazil appear distinct from other Brazil samples [7], while a single genome from Rondônia in Western Brazil–bordering Bolivia–clustered with the Peru/Bolivia/Colombia isolates (**Fig 5C, lower right**). These data support the hypothesis that geography influences population genetic structure in *L. braziliensis*. Upon closer examination of the dense cluster of highly similar genome sequences from Northeastern Brazil (**Fig 5C, inset**), we observed a separation between SWGA sequences from Bahia (inset; triangles) and those from Pernambuco (inset; circles). To confirm that this separation was not an artifact of using SWGA, we included two control samples in which genome sequence data was generated from the same cultured laboratory clone of *L. braziliensis* from Brazil by either traditional WGS (**Fig 5C, inset; black circle**) or SWGA (**white triangle**). These two data points are indistinguishable from each other on PCA and cluster with other genomes from Brazil, demonstrating that the SWGA method itself is not likely to be a significant contributor to the variation observed in this analysis.

To view the genomic variation for these 59 genomes with more clarity, we plotted the first four principal components–which collectively account for over 38% of the total variance–separately, allowing us to see how each sample contributes to each principal component (**Fig 5D**). When viewed in this way, PC1 clearly separates two of the Colombia isolates from all other genomes, consistent with a high number of SNPs previously described for these samples [8]. PC2 separates Brazil samples from all other samples, regardless of whether they are from SWGA or WGS of cultured isolates. PC3 separates the two WGS samples from Paudalho, Pernambuco, Brazil, from all others, while PC4 separates samples originating from Colombia versus Peru. Collectively, these data point to country and, to a much smaller extent regional differences, as being associated with genetic variation in *L. braziliensis*. Our data show that integrating these data opens the door to comparing SWGA data in the context of a growing number of WGS datasets for *L. braziliensis*.

Phylogenetic analysis supports the hypothesis that both the forested Pernambuco, Brazil samples and two samples from Colombia are quite unique (**Fig 5E**). Like the PCA, this tree shows that the single sample from Western Brazil is more similar to samples from Peru and Bolivia. The SWGA samples form a monophyletic clade with previously published genomes also from Bahia, Brazil, and are closely related to the non-forest Pernambuco, Brazil samples. Since we integrated our SWGA genomes with published WGS genomes, we wanted to rule out a potential issue in which uneven coverage from SWGA, but not WGS, could contribute to the structure observed in our phylogenetic tree. To evaluate the robustness of the tree to loss of signal, we dropped 20%, 40%, or 60% of the SNPs, each time regenerating the tree (**S3 Fig**). We found that > 40% of the SNPs must be removed before our SWGA samples no longer form a monophyletic clade with the published Bahia genomes generated by WGS, and instead become sister clades. Our phylogenetic analysis further supports the conclusion that SWGA and WGS genomes can be compared, since our cultured laboratory clone of *L. braziliensis* falls within the same clade with extremely short branch lengths and a high bootstrap value.

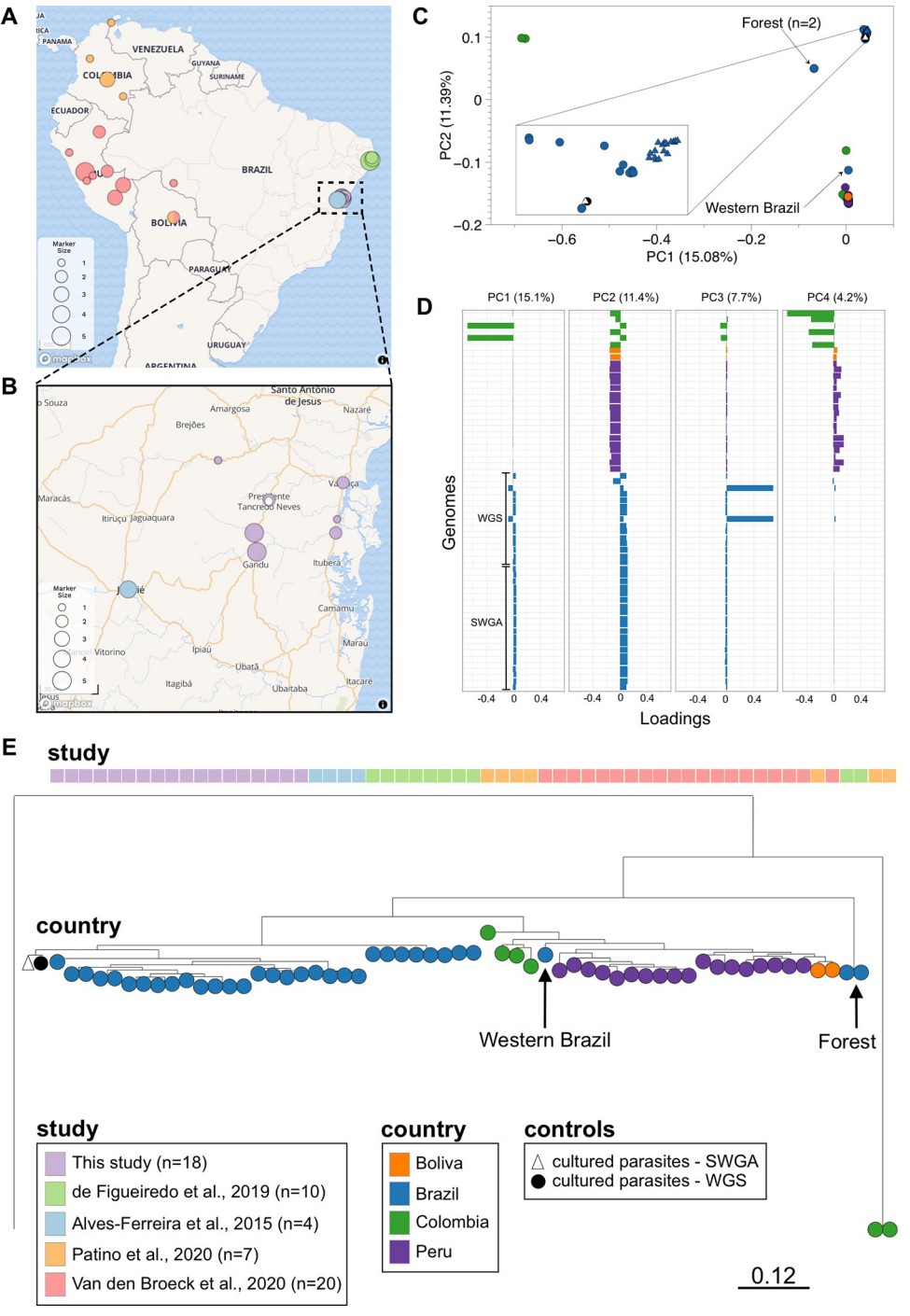

**Fig 5. Integrating SWGA and WGS genomes for population genomics.** (A) Map showing all 59 samples, from this study and four previously published reports, included in the analysis [7, 46, 8, 47]. (B) Zoomed in view of Bahia, Brazil showing region covered by samples from this study. White point indicates position of field hospital where patients were seen. (C-D) Principal component analysis of SNP data from 59 genomes, colored by country of origin. (E) Maximum likelihood tree constructed using 877713 variants from 59 *L. braziliensis* genomes and the *L. guyanensis* outgroup, compared to the *L. braziliensis* reference. Branch length of outgroup was shortened for figure preparation. Tree is rooted using the *L. guyanensis* outgroup. The same cultured laboratory clone of *L. braziliensis* from Brazil was sequenced either by traditional WGS **(black circle)** or SWGA **(white triangle)**. Map data from Maps Mapbox (www.mapbox.com/about/maps) and OpenStreetMap (www.openstreetmap.org/about).

## Identifying variants unique to Northeast Brazil where treatment failure rates are high

None of the previously published *L. braziliensis* genomes have reported treatment outcome for patients from which isolates were generated. Thus, our SWGA genomes are the only ones with available treatment outcome data, leaving us underpowered to test for parasite polymorphisms linked to treatment outcome. However, treatment failure rates are reportedly high in Northeastern (NE) Brazil [48] and our SWGA genomes cover a region in NE Brazil not well represented by previous WGS studies. This, together with the fact that many of our SWGA genomes (15/18) came from patients who failed therapy with pentavalent antimony (S2 Table), prompted us to ask whether our data could be used in a proof-of-concept exercise to identify parasite variants unique to NE Brazil and, therefore, potentially linked to treatment failure. Toward this end, we carried out a systematic identification and annotation of genomic variants from all 59 *L. braziliensis* genomes available, yielding over 600,000 high-quality variants, including nearly 110,000 missense and 634 frame-shift variants (Fig 6A, column labeled 'S. America (total)'). Over 120,000 of these variants were present in our SWGA genomes (Fig 6A, column labeled 'SWGA (total)'), and our data identified 5,812 novel variants not previously observed in other studies including 1,204 missense and 277 frame-shift variants (Fig 6A, column labeled 'SWGA (new)'). Notably, nearly half of the total *L. braziliensis* frame-shift

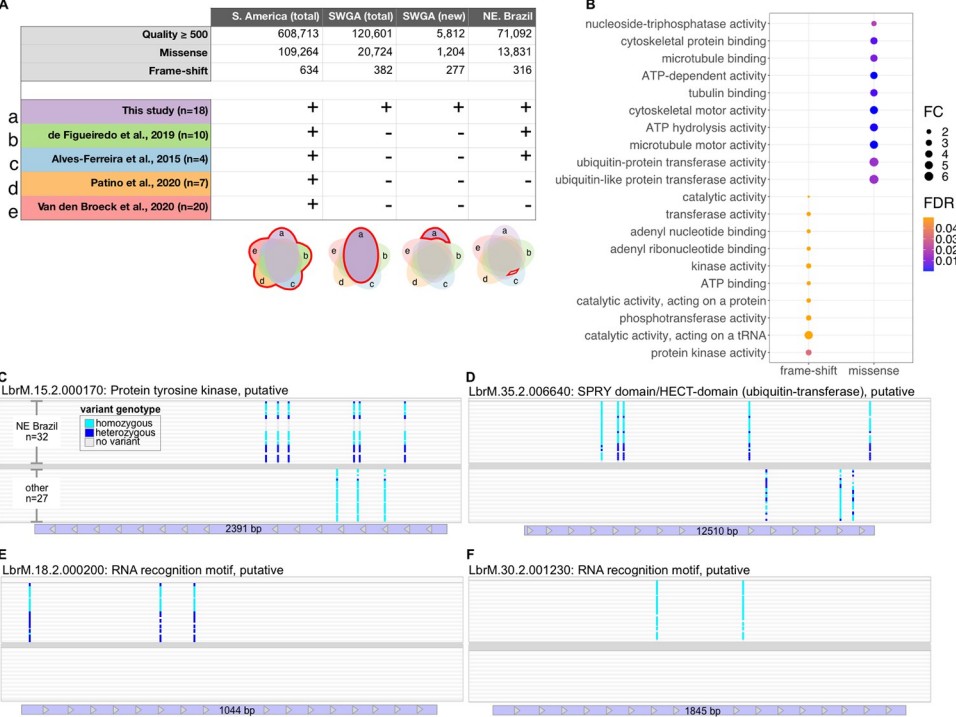

**Fig 6. Identification of variants unique to Northeastern Brazil.** (A) Table showing variants identified by integrated analysis of WGS and SWGA genomes (top), and studies included (+) or excluded (-) from the analysis (bottom). Venn diagrams indicate how each of the five studies (labeled a-d; [7, 46, 8, 47]) were used in the integrated analysis to generate the variants shown in table column above. (B) Bubble chart showing results of Gene Ontology (GO) enrichment for Molecular Function terms associated with 149 genes containing frame-shift variants (left) or 152 genes identified with high-frequency missense mutations in Northeast (NE) Brazil (right). All terms shown were associated with ≥ 5 genes. FC = fold change; FDR = false discovery rate (Benjamini-Hochberg correction). (C) Four representative parasite genes that were enriched for high-frequency missense mutations in genomes from Northeast Brazil.

mutations were contributed by our SWGA data. To confirm that the relatively large number of frame-shift mutations observed in our experiments was not an artifact of SWGA, we examined the INDEL to SNP ratio for 4 SWGA genomes from Bahia, 4 WGS genomes from Bahia, 4 WGS genomes from Pernambuco, and 4 WGS genomes from Peru. We found the INDEL to SNP ratio to be 0.256, 0.205, 0.217, and 0.179, respectively. Similarly, when we carried out WGS and SWGA on pure cultured parasites (**Fig 5**) we measured INDEL to SNP ratios of 0.251 and 0.219, respectively. Taken together, these data suggest a generally higher trend for this ratio in genomes from NE Brazil, rather than a technical issue that results in more INDELs in SWGA genomes.

Next, we focused our analysis on frame-shift and missense mutations, since they have a high potential for impacting protein sequence. In particular, we were interested in these variants when present in NE Brazil–including our 18 SWGA genomes and 14 genomes from two other studies [7,46]–but absent from genomes collected from Colombia, Bolivia, Western Brazil, and Peru (**Fig 6A,** column labeled 'NE Brazil'). 316 frame-shift mutations were found to be specific to NE Brazil and occurred in 303 genes, of which 51% (154) were annotated as conserved hypothetical genes (**S3 Table**). Gene ontology enrichment analysis of the remaining 149 genes revealed enrichment of functional terms associated with post-translational modifications, including protein phosphorylation (2.7-fold enrichment; FDR = 0.03) (**Fig 6B, left**). In addition, we identified 13,831 missense mutations specific to NE Brazil. We reasoned that many of these variants were likely observed at low frequency (only found in one or a few samples), thus we further refined this list by selecting for variants that were observed at high frequency in NE Brazil but not elsewhere (see methods). This analysis yielded 1916 variants. To focus on genes with the potential to be most impacted by these mutations, we selected only genes that had $\geq$ 2 of these missense mutations, resulting in a list of 347 genes, of which 52% (195) were conserved hypothetical proteins (**S3 Table**). GO analysis of the remaining 152 genes showed significant enrichment of ubiquitin transferase activity ($>$ 6-fold enrichment; FDR = 0.01) (**Fig 6B, right).** Included amongst this list were 12 genes with putative kinase domains, 4 SPRY-domain/HECT-domain-containing (ubiquitin-transferase) proteins (LbrM.32.2.004170, LbrM.13.2.001230, LbrM.07.2.000290, and LbrM.35.2.006640), one ubiquitin carboxyl-terminal hydrolase (LbrM.16.2.000720), one putative E1 ubiquitin-activating enzyme (LbrM.34.2.002970), and two putative cullin protein neddylation domain-containing proteins (LbrM.16.2.001260 and LbrM.25.2.001240) (**S3 Table**). Five RNA binding proteins were also identified in this analysis (LbrM.18.2.000200, LbrM.18.2.001450, LbrM.24.2.001860, LbrM.29.2.001510, LbrM.30.2.001230, and LbrM.33.2.001710). In some cases, these genes had high frequency variants both within and outside of NE Brazil, but present at different locations in the gene (**Fig 6C and 6D**). For other genes, high frequency missense mutations were only observed in NE Brazil (**Fig 6E and 6F**). Collectively, these proof-of-concept results underscore the potential for SWGA to allow researchers to link genetic polymorphisms in *Leishmania* with experimental covariates and raise the possibility that *L. braziliensis* strains circulating in NE Brazil may undergo unique post-transcriptional or post-translational modifications.

## Discussion

The slow growth of *L. braziliensis*, combined with low parasite burden present at the site of the lesion and relative scarcity of infrastructure to support high-throughput sequencing in areas endemic for CL, have made it difficult to isolate, culture, and sequence a diverse range of parasite strains for population genomic studies. One recent strategy for addressing these challenges in *L. donovani* used custom biotinylated 'bait' sequences and streptavidin-conjugated beads (Agilent SureSelect technology) to enrich for parasite DNA in samples from visceral

leishmaniasis patients [15]. This method may have several advantages over SWGA. For example, SureSelect is likely to be more sensitive than SWGA and can amplify parasite gnomes from samples with as low as 0.006% *Leishmania* DNA [15]. This could be particularly important for generating parasite genomes from patients with mucosal leishmaniasis, or from patients that cure following a single round of antimony treatment, as both these patient populations tend to have very low parasite burden. Although it remains to be tested, in theory the efficiency of SureSelect technology should not be impacted by host background. In contrast, SWGA primers will likely need to be redesigned for host backgrounds that different significantly from human (e.g., sandfly vector). However, the relatively high cost of SureSelect assays coupled with the need to redesign new baits for different species of *Leishmania* and the high amount of input DNA required (minimum 100 ng) limit more widespread adoption of this approach. The data presented here show that simple pools consisting of ten 8-mer primers can be used to selectively amplify *L. braziliensis* genomes–and likely *L. major*–from complex primary patient samples. Aside from these oligonucleotide primers, only the Phi29 polymerase is needed and the SWGA proceeds as an isothermal room-temperature reaction, bypassing the need for a thermocycler. Since SWGA is an amplification-based protocol, only small amounts of total DNA (as low as 5ng) are needed. Taken together, our data show that SWGA is a low-cost and easily scalable method to generate high resolution population genomic data from *Leishmania* species, even in resource-limited areas.

Although we successfully amplified 18 parasite genomes from primary patient samples, this represented only a 27% success rate from the 66 samples we attempted to amplify with SWGA. One open question is how the efficiency of the SWGA method can be improved so that a higher number of patient samples yield parasite genomes. Host-specific restriction enzymes [20,49] may offer one appealing solution for *Leishmania*, particularly since *L. donovani* reportedly lacks C-5 DNA methylation, potentially opening the doors to using methylation-sensitive restriction enzymes to preferentially degrade host DNA [50]. Based on our data from SWGA of synthetic controls (**Fig 1C**), primer sets 1 and 4 yielded the greatest percent of reads aligning to *L. braziliensis*, while primer sets 2 and 3 performed more poorly. Interestingly, sets 1 and 4 share more primers in common with each other, than they do with sets 2 and 3 (**S1 Table**). Thus, we could use the sequences in sets 1 and 4 to refine the SWGA algorithm to identify new primers that may demonstrate improved performance. Despite these limitations, SWGA offers several exciting potential uses for *Leishmania* genetics. The Phi29 polymerase used in SWGA is highly processive and can produce amplicons up to 100kb or more in length, potentially allowing long-read sequencing of SWGA reactions to resolve complex regions in the parasite genome. We expect that SWGA will make capturing genomes of *Leishmania* parasites from sympatric mammalian hosts (e.g. human and canine) and insect vectors all from the same geographic area relatively straightforward, thus empowering the design of sophisticated population genetic studies.

Our 18 SWGA genomes included 15 from patients who failed treatment after a single round of chemotherapy with antimony (**S2 Table**). This bias in favor of successful SWGA of parasite genomes from patients who fail therapy is likely due to the higher parasite burden observed in these patients [10], thus putting the total amount of parasite DNA above a threshold for successful SWGA. Understanding why some patients have higher parasite load than others–prior to initiating chemotherapy–may help identify the root causes of treatment failure in this disease. There are many possible explanations, including variable parasite load in the insect vector, variability in host immunity, differential host immune evasion by the parasite, differing parasite replication rates, and more. All these potential explanations could involve parasite strain genetics, yet prior to this study little was known about how *L. braziliensis* strains in NE Brazil where failure rates are high, compared to those observed elsewhere in South

America. Future studies to formally identify parasite variants associated with treatment outcome will require some consideration for how to successfully obtain genomes from patients who cure and, therefore, have the lowest parasite load prior to treatment. One potential solution would be to perform SWGA on skin biopsies collected from patients early in the course of disease, before the development of an ulcer. Previous studies have shown that this early stage of the disease is when parasite burden and failure rates are highest [51]. Notably, our screening approach (**Fig 4A–4E**), is scalable and could be used to tackle this challenge by rapidly testing many different samples and patients to identify the optimal setting to generate genomes from very low burden infections.

By integrating our SWGA genomes with public WGS data, we were able to carry out a population genetic study of *L. braziliensis* that spanned four S. American countries. As a proof-of-concept exercise, we identified variants unique to NE Brazil thus highlighting the feasibility of using SWGA data for genetic association studies in *L. braziliensis*. The high-frequency variants we identified in NE Brazil were enriched in protein kinases, RNA-binding proteins, and ubiquitin-transferases. We hypothesize that these mutations may impact RNA or protein stability in the parasites. Interestingly, *Leishmania* and *Trypanosoma* parasites lack traditional promoter-based gene regulation and thus rely heavily on post-transcriptional and post-translational mechanisms for modulating gene expression in the face of environmental stressors and cues [52,53]. For example, RNA binding proteins in *Trypanosomes* are critical for differentiation of the parasite through its lifecycle [54,55]. Collectively, our data underscore the potential for SWGA to be used in population genomic studies to identify parasite genetic polymorphisms linked to experimental covariates.

## Materials and methods

### Ethics statement

This study was conducted according to the principles specified in the Declaration of Helsinki and under local ethical guidelines (Ethical Committee of the Maternidade Climerio de Oliveira, Salvador, Bahia, Brazil; and the University of Pennsylvania Institutional Review Board). This study was approved by the Ethical Committee of the Federal University of Bahia (Salvador, Bahia, Brazil) and the University of Pennsylvania IRB (Philadelphia, PA; protocol #834504). All patients provided written informed consent for the collection of samples and subsequent analysis. All animal work was carried out in accordance with the recommendations in the Guide for the Care and Use of Laboratory Animals of the National Institutes of Health. The protocol was approved by the Institutional Animal Care and Use Committee, University of Pennsylvania.

### Human and mouse sample collection

4-mm diagnostic skin lesion biopsies were collected prior to initiating treatment from the border of the lesion of CL patients, and DNA was extracted using the Wizard Genomic DNA Purification Kit (Promega). CL diagnosis was determined by a positive skin lesion PCR for *L. braziliensis* and a positive intradermal skin test with *Leishmania* antigen. These diagnostic DNA samples were the same ones used in this study. For some patients, an additional biopsy was collected and stored in RNAlater (Thermo Fisher Scientific) for shipment. Biopsies were homogenized, and DNA was extracted using the MP Bio FastPrep Tissue Homogenizer and Qiagen Blood and Tissue kit according to the manufacturer's instructions.

For mouse experiments, *L. braziliensis* (MHOM/BR/01/BA788 strain) and *L. major* (Friedlin strain) parasites were grown in Schneider's insect medium (GIBCO) supplemented with 20% heat-inactivated fetal bovine serum (Atlanta Biologicals) and 2 mM glutamine (Sigma).

Metacyclic promastigotes were enriched from stationary-phase parasite cultures by density gradient centrifugation before infection as previously described [56]. Briefly, parasites were suspended in PBS and layered on a step gradient of 40% and 12% Ficoll 400 (Sigma) before centrifuging at 2400 rpm for 10 minutes. C57BL/6 mice were infected intradermally in the ear with 1x10$^6$ *L. braziliensis* or *L. major*. At the peak of ear swelling (~4–6 weeks post-infection), mice were humanely euthanized, ears were collected, homogenized, and DNA extracted as described above for human samples.

## SWGA primer design and validation

We used the program swga [16] to generate a list of 172 candidate primers that preferentially bind to the *Leishmania braziliensis* reference genome (MHOM/BR/75/M2904 2019) over a complex background genome that consisted of human (GCA_000001405.28), *Staphylococcus aureus* (GCA_000746505.1), and *Streptococcus pyogenes* (GCA_000006785.2). We scored these candidate primers and designed primer sets using an updated machine-learning-guided and thermodynamically-principled version of the SWGA algorithm, swga2.0 [31](software available at https://anaconda.org/janedwivedi/soapswga). Overall, 23 unique 8-mer primers with the highest evaluation scores calculated from swga2.0 were generated (Integrated DNA Technologies). The last two bases of the primers were phosphorothioated, which prevents primer degradation by phi29 polymerase [21]. *In silico* validation was carried out by counting exact matches for each SWGA primer against a range of target and background genomes using the Unix *grep* command, and hits per Mbp and the fold difference in predicted binding sites were calculated and visualized using Prism 9. The target genomes included *L. braziliensis* (see above), *L. major* (TriTrypDB-55_LmajorFriedlin), *L. donovani* (TriTrypDB-46_LdonovaniBPK282A1), *L. infantum* (TriTrypDB-56_LinfantumJPCM5), and *L. amazonensis* (TriTrypDB-56_LamazonensisMHOMBR71973M2269). Background genomes included human (Homo_sapiens.GRCh38), *Mus musculus* (GCF_000001635.27_GRCm39), and *Canis lupus familiaris* (Canis_lupus_familiaris.CanFam3.1). The human, mouse, and canine reference genomes were filtered to only include the autosomal chromosomes, sex chromosomes, and mitochondrial DNA for the analysis. Primers were grouped into four sets of 10 primers each (**S1 Table**). FastQ Screen [57] (sampling 10$^5$ reads per sample) was used to assess the selectivity of the SWGA primers on 9 pre-and post-SWGA samples against a panel of different reference genomes including human, *S. aureus*, *S. pyogenes*, *L. braziliensis* and the *L. braziliensis* maxicircle. Proportion of reads mapping to each reference genome were visualized using Prism 9. Genomic DNA extracted from human foreskin fibroblasts (HFF) cells and an axenic culture of *L. braziliensis* promastigotes using the DNeasy Blood and Tissue kit (Qiagen) were mixed to generate 1% and 0.1% *L. braziliensis*:human DNA (w/w).

## SWGA on primary patient and mouse samples

DNA from human or mouse samples was quantified using a Qubit 3.0 fluorometer. qPCR was performed on a ViiA 7 machine (Applied Biosciences) using SsoAdvanced Universal Probes Supermix (BioRad) for both *Leishmania* kinetoplast DNA [45] and the human 18S rRNA gene (Biomeme Inc). Ct values for *Leishmania* were normalized using the human 18S rRNA gene to prioritize lesions with the highest parasite burden for SWGA. All qPCR reactions were carried out in duplicate. SWGA was performed by combining ~50 ng of the sample DNA, 3.5mM of an SWGA primer set, 1x phi29 buffer, 30 U of phi29 polymerase enzyme (New England Biolabs), 4mM dNTPs (Thermo Fisher Scientific), 1% bovine serum albumin and nuclease-free water in a total volume of 50μL. Thermocycler cycling conditions included a 1 hr ramp down step (35˚C to 30˚C; 10 min per degree), 16 hr amplification

step at 30˚C, 10 min denaturing step at 65˚C and hold at 4˚C. Of the four primer sets reported here, PS1 and PS4 performed best, thus PS2 and PS3 were only used in a second round of SWGA following amplification with PS1 and PS4. For second-round SWGA reactions, ~50 ng of first-round SWGA product was subjected to a second round of SWGA with a different primer set. Ten ng of first-round or second-round SWGA product was used to generate libraries using the Hackflex [58] protocol and subjected to shallow sequencing on an Illumina NextSeq 500 or NextSeq 2000 to produce 1–4 million 75 or 150 single-end reads per SWGA reaction. Reads were trimmed with Trimmomatic [59], aligned to the appropriate *Leishmania* reference genome using Bowtie2 [60], and summarized with MultiQC [61]. All SWGA reactions that showed >20% reads aligning to *L. braziliensis* were pooled by patient and subsequently resequenced to generate ≥100 million paired-end 150 bp reads. Genome coverage was estimated based on the median gene coverage, excluding genes with outlier coverage, removed with iterative Grubbs' test.

## Variant calling, phylogeny, and somy analysis

Sequencing data from different SWGA primer sets were combined for each sample using the Unix *cat* command. In addition to data from the 18 SWGA samples, publicly available raw sequence reads were also obtained for 41 *L. braziliensis* cultured isolates from Colombia, Bolivia, Brazil, and Peru [7,8,46,47] that were subjected to whole genome sequencing (WGS). Reads were trimmed with Trimmomatic [59] (filtering parameters: LEADING:3 TRAILING:3 SLIDINGWINDOW:4:15 MINLEN:36) and mapped to the *L. braziliensis* MHOM/BR/75/M2904 2019 reference genome using bwa-mem v.0.7.17 [62]. Alignments were reported in bam files, which were sorted, and indexed with SAMtools [63], and reads were tagged with a sample ID using Picard Tools *AddOrReplaceReadGroups* [64] similar to previously described [65]. Genome coverage was estimated using BEDtools *genomecov* command with 100 bp windows [66]. The percent of the *L. braziliensis* genome covered at ≥1x, 5x, and 10x was calculated from the resulting bed file. SNPs and indels were called using The Genome Analysis Toolkit (GATK) v.4.1.0.0 [67] *HaplotypeCaller* and Freebayes v.1.3.2 [68] in 'discovery' mode, with a minimum alternative allele count set to ≥5. Only variants found by both methods were retained for downstream analysis. The SWGA and WGS data were merged and sorted with BCFtools v.1.9 [69] and regenotyped using Freebayes. A bed file that contained only regions with ≥10x coverage in at least 14 out of the 18 SWGA samples was used to filter the SWGA and public WGS data for population and phylogenetic analysis.

For phylogenetic analysis, biallelic sites were selected with BCFtools [69], and variant calls were filtered by quality (QUAL>500) with VCFtools [70] and by linkage disequilibrium with Plink v.1.9 [71] (parameters used: $r^2$ = 0.5, step size = 1, window size = 10kb). Principal component analysis was carried out with Plink v.1.9. For phylogenetic tree generation, *L. guyanensis* MCAN/CO/1985/CL-085 (ERR205773) was mapped to the *L. braziliensis* reference as above to be used as an outgroup to root the tree. Sequences were extracted from the merged SWGA, public, and outgroup variant call format (VCF) file with vcf2phylip v2.8 [72] and a maximum likelihood phylogenetic analysis was performed using IQ-TREE v.2.0.6 [73] (parameters used: *ModelFinder Plus*, and 10000 bootstrap replicates for SH-aLRT). The resulting tree and geospatial data were visualized with Microreact [74]. To investigate the potential for lower breadth of coverage with SWGA to impact our phylogenetic analysis, we divided genomes into 10 kb segments and removed 20%, 40%, and 60% of the segments using BEDtools and BCFtools. The filtered VCF file went through the same workflow as above. Genomic variants were annotated with snpEff [75], which was configured using a custom database

prepared from the *L. braziliensis* genome fasta file, coding sequence (CDS) fasta file, Gene Transfer Format (GFF) file, and codon usage data, all of which were obtained from TriTypDB. org (release 58) [76,77]. Filtering of variants by quality and type was carried out using SnpSift [78], and comparisons of variants between any two sets of samples were carried out using the *isec* function from BCFtools [69]. For high-frequency variants, snpSift was used to identify only missense mutations with an allele count greater than the number of samples in the group (n = 32 for NE Brazil, n = 27 for non-NE Brazil). For example, since *Leishmania* is diploid, an allele count of 32 in a group of 32 samples could be achieved if all samples were heterozygous for a mutant allele or if half of the samples were homozygous. Data visualization was carried out using R/Bioconductor [79,80], the vcfR package [81], ggplot2 [82], DataGraph v4.7.1, Prism 9, and Sketch v91. Chromosomal somy estimation was based on the proportion of reads in the alternate allele in biallelic heterozygous positions. VCF files were imported in R using vcfR and only biallelic positions were kept that had at least 10 reads in each allele and a total read depth of at least 30 and lower than 200. For each chromosome, the proportion of reads corresponding to the alternate allele in each SNP position was obtained and their distribution was used to infer the chromosomal somy.

## Supporting information

**S1 Fig. Specificity of SWGA primers for the *L. braziliensis* nuclear genome.** Plot showing number of reads (out of 100,000 subsampled reads) from each of 9 patient samples (points) that mapped to genomes of human, *L. braziliensis*, *Staphylococcus aureus*, *Streptococcus pyogenes*, and the *L. braziliensis* kinetoplast maxicircle.
(TIFF)

**S2 Fig. Genome-wide coverage by SWGA.** Coverage plots for 35 *L. braziliensis* chromosomes in SWGA data from a single patient (#7; blue lines) compared to whole genome sequencing (WGS) of pure, cultured *L. braziliensis* (orange lines). Data were merged from all SWGA primer sets to maximize coverage.
(TIFF)

**S3 Fig. SWGA of *L. braziliensis* results in robust phylogenetic clustering.** (A) PCA plots (B) red inset from PCA plots shown in panel A, and (C) maximum likelihood phylogenetic trees showing relationship between our SWGA genomes (triangles) and previously published WGS samples (circles), all from Bahia, Brazil. Trees shown in panel C are focused on only the left-hand region of the tree shown in Fig 5E. White triangle and black circle indicate SWGA and WGS from cultured parasites, respectively. Red branches indicate monophyletic group.
(JPG)

**S1 Table. SWGA primers.** Sequences and set assignments for each of the 23 SWGA primers used in this study.
(XLSX)

**S2 Table. Patient details.** Clinical and demographic data for cutaneous leishmaniasis patients that participated in this study.
(XLSX)

**S3 Table. *L. braziliensis* genomic variants.** Frame-shift and missense variants enriched in *L. braziliensis* parasites present in Northeast Brazil.
(XLSX)

## Author Contributions

**Conceptualization:** Olivia A. Pilling, Alexander S. F. Berry, Dustin Brisson, Daniel C. Jeffares, Daniel P. Beiting.

**Data curation:** Olivia A. Pilling, João L. Reis-Cunha, Clara R. Malekshahi, Elise Krespan, Daniel P. Beiting.

**Formal analysis:** Olivia A. Pilling, João L. Reis-Cunha, Cooper A. Grace, Daniel P. Beiting.

**Funding acquisition:** Phillip Scott, Daniel P. Beiting.

**Investigation:** Christina K. Go, Cláudia Lombana, Camila F. Amorim, Alexsandro S. Lago, Lucas P. Carvalho, Edgar M. Carvalho, Phillip Scott, Daniel C. Jeffares.

**Methodology:** Olivia A. Pilling, João L. Reis-Cunha, Cooper A. Grace, Clara R. Malekshahi, Elise Krespan.

**Project administration:** Clara R. Malekshahi, Elise Krespan, Daniel P. Beiting.

**Resources:** Phillip Scott.

**Software:** João L. Reis-Cunha, Cooper A. Grace, Matthew W. Mitchell, Jane A. Yu, Yun S. Song, Dustin Brisson.

**Supervision:** Daniel C. Jeffares, Daniel P. Beiting.

**Visualization:** João L. Reis-Cunha, Cooper A. Grace, Daniel P. Beiting.

**Writing – original draft:** Olivia A. Pilling, Daniel P. Beiting.

**Writing – review & editing:** Olivia A. Pilling, João L. Reis-Cunha, Cooper A. Grace, Alexander S. F. Berry, Matthew W. Mitchell, Edgar M. Carvalho, Dustin Brisson, Phillip Scott, Daniel P. Beiting.

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
