## [Decision Letter · Decision Letter 0]

8 Nov 2022

Dear Dr Beiting,

Thank you very much for submitting your manuscript "Selective whole-genome amplification reveals population genetics of Leishmania braziliensis directly from patient skin biopsies" for consideration at PLOS Pathogens. As with all papers reviewed by the journal, your manuscript was reviewed by members of the editorial board and by several independent reviewers. In light of the reviews (below this email), we would like to invite the resubmission of a significantly-revised version that takes into account the reviewers' comments. They appreciate the attention brought to an important problem but raise some concerns about the manuscript in its current state, which must be addressed before we could consider a revised version of your manuscript. We therefore request that you modify the manuscript according to reviewers’ recommendations before we can consider it for acceptance.

We are returning your manuscript with two reviews. Overall, the two reviewers came to similar conclusions about the paper. After reading these reviews and the manuscript carefully, we recommend a Major Revision based on these critiques. We cannot promise publication after receipt of a revised manuscript, of course. However, we are looking forward to receiving your revision, and with significant work, the manuscript should be suitable for resubmission, if you choose to do so. Your revisions must address the specific points made by each reviewer.

In particular, as the reviewers indicate, the impact of this work would be significantly increased by analyzing any correlations between genetic profiles provided and treatment outcome. This should be done by 1) comparing/contrasting sequencing results documented in the manuscript for the strains/biopsies linked to treatment success or failure to existing sequencing data from available genomes (from more susceptible or more resistant strains) and 2) validating susceptibility results in vitro for a subset of identified clinically resistant or susceptible variants. If sample size was not sufficiently powered to draw strong conclusions about relationships between variants and outcome/resistance (as one reviewer suggests), this section could be presented as a proof-of-concept exercise to demonstrate the feasibility of applying SWGA to studies associating clinical phenotypes and parasite genetics.

In addition, please pay attention to the following reviewer suggestions and give them your due consideration:

Reviewer 1:

-Provide a flow chart of samples and inclusion/exclusion processes for analysis of outcomes.

-More clearly highlight clinical and demographic data for clinical isolates as well as patient biopsies.

-Indicate the coverage of selectively-amplified DNA for bacterial commensals.

Reviewer 2:

-Further discuss the limitations of SWGA compared to Sure Select arrays.

-Identify the causes and consequences of the coverage fluctuations seen across chromosomes (Figure 2D).

-Address the indicated concerns for Figures 3 and 4.

-Discuss whether SWGA may affect clustering (Figure 5) or introduce mutations (Figure 6).

The full comments from the reviewers follow this cover letter.

We cannot make any decision about publication until we have seen the revised manuscript and your response to the reviewers' comments. Your revised manuscript is also likely to be sent to reviewers for further evaluation.

Sincerely,

Dawn Marie Wetzel

Guest Editor

PLOS Pathogens

Margaret Phillips

Section Editor

PLOS Pathogens

Kasturi Haldar

Editor-in-Chief

PLOS Pathogens

orcid.org/0000-0001-5065-158X

Michael Malim

Editor-in-Chief

PLOS Pathogens

orcid.org/0000-0002-7699-2064

Reviewer's Responses to Questions

**Part I - Summary**

Reviewer #1: In the manuscript entitled Selective whole-genome amplification reveals population genetics of Leishmania braziliensis directly from patient skin biopsies, Pilling et.al., present a very interesting, useful and feasible method for selective genome wide amplification of Leishmania DNA from complex tissue samples. Overall the paper is well written, although quite long and suggest some text reduction especially in the results section. My major comment relates to the relationships ascertained for variants and therapeutic outcome as detailed below.

Reviewer #2: In the manuscript ‘Selective whole-genome amplification reveals population genetics of Leishmania braziliensis directly from patient skin biopsies’, Pilling et al for the first time apply selective whole genome amplification (SWGA) on skin biopsies from Brazilian patients infected with Leishmania braziliensis. The manuscript is largely a proof-of-principle study that SWGA can be also applied to Leishmania, similar to its application to many other pathogens. The first three figures of this manuscript are dedicated to critically assess this method and show that SWGA (i) allows for over 60-fold enrichment of parasite over host DNA in synthetic, ‘spiked’ samples, with the same primer set working across all major parasite species, (ii) allows to enrich parasite DNA from samples obtained from infected mice and patients attaining a 10x coverage for 80% of the genome, and (iii) allow to monitor somy variation in spiked samples while being less resolutive on clinical material. They then apply the method on archived samples, a series of newly sequenced genomes and available genomes published by others to (i) gain insight into structural genomic changes (SNP, ndels, frameshifts), (ii) carry out a population genomic analysis using a phylogenetic approach, and (iii) reveal unique, regional variants for L. braziliensis in Brazil. The paper is well written, and the results are well controlled and clearly presented, making a convincing case that SWGA will be useful for the Leishmania community. Enthusiasm is somewhat tempered by the limitation of this method caused by inherent, technical noise that sees to preclude genome analysis in a more quantitative way, such as changes in SNP frequency, in chromosome and gene copy number, which represents key information for GWAS and biomarker discovery. Nonetheless, the method is a powerful tool for the assessment of unique structural features of a given genome and to use this information to assess the evolutionary relationship between genomes.

Major comments:

Discussion: While the advantages of SWGA over the more cost-intensive, commercially available solutions are convincingly presented throughout the manuscript, a more critical discussion of the limitations of SWGA compared to Sure Select arrays should be added. This would also allow to better highlight the strengths and applications of SWGA. How are the methods comparing with respect to technical background and quantitative genome assessment on for example gene and chromosome CNVs, or variations in SNP frequencies?

Figure 2: Panel 2B should be either discussed at the beginning (it is discussed at the very end of this chapter) or the panel labelling and organization should be adapted and come at last. The data shown in panel 2D should be more critically assessed and the cause (technical) and consequences (no quantitative genetic information possible) of the important coverage fluctuations seen across the entire chromosomes outside the ‘complicated’ areas should be discussed.

Figure 3: There is a misconception here: a SNP frequency ratio of 0.5 is not proof of a disomic state! Here, populations are sequenced and thus the somie values of each individual parasite are integrated in the final genome sequence. Thus, if the trisomy is caused by either of the two haplotypes, the heterozygous SNPs will be balanced again at 0.5 in the population. Also, for a meaningful analysis of SNP frequencies, a robust assessment of the read depth needs to be available, which is not the case for SWGA. I would remove this figure or add it to supp data, indicating that SWGA is less useful in estimating somie differences based on either read depth variations or SNP frequency shifts.

Figure 4: How does the nested procedure – i.e. a first shallow sequencing followed by sequencing of pooled libraries – affect SNP frequencies and read depth (see comments above)? Shouldn’t this introduce a major bias given that the initially obtained 20% includes both unique (hence the pooling), but also overlapping regions that will then be over-represented in the second round of sequencing? How does this affect the more quantitative aspects of comparative genome analyses, which are at the core of biomarker discovery? These drawbacks need to be discussed.

Figure 5 panel E: Define the triangle and the black dot in the legend. Based on the color code, all SWGA samples are clustering together. This is not properly highlighted in the results section. This signal should be pointed out and a role (even if minor) of SWGA on clustering should be discussed.

Figure 6: The functional enrichment observed for the structural mutants that change the protein sequence changes is potentially very interesting! The fact that half of the unique variants are contributed by the SWGA samples is a bit concerning – how sure is it that these are not introduced by the method? It is unclear to me how many samples show mutations in a given gene (same position or different), which would indicate convergence and thus selection. Have any of these genes been previously linked to treatment failure? I am surprised that these changes were not correlated to the observed and documented outcome of treatment across the samples with available genomes (including the 59 genomes sequenced by others). Is there any correlation, even if statistically insignificant?

**Part II – Major Issues: Key Experiments Required for Acceptance**

Reviewer #1: 1) It is very difficult to follow the description of the results section on SNP analyses in the context of therapeutic failures. The mix of samples and the lack of clarity of exclusion of some of them makes it hard to define the extent to which conclusions in relation to outcome could be drawn. As a first step I would request a flow chart of samples, and the process of inclusion/exclusion for analysis regarding outcome. Then the clinical and demographic metadata should be clearly defined from isolates as well as biopsies, so that the reader can contrast the quality of the definitions of outcome and therefore the interpretability of the data.

2) If any relationships are to be drawn in relation to susceptibility, at least some of the variants should be validated at least in vitro, since the authors are talking about “parasite drug resistance”

3) I strongly suggest that the scope of the paper be the method and the potential applicability, more than the relationship of variants with outcome and/or resistance. The latter seems to stem from a secondary analysis of the data that is not really powered to draw the expected conclusions, and could be better presented as an “academic” exercise of feasibility of implementing this method to association studies of clinical phenotypes and parasite genetics.

4) It would be interesting to show coverage of the selectively amplified DNA for bacterial commensals to show that this amplification tool is really specific to Leishmania DNA and that the use of S. aureus genome as background was indeed useful and successful.

5) Not clear why the somy analyses from clinical samples were not as conclusive as isolated parasites. How can high host DNA content affect somy analyses?

Reviewer #2: In the manuscript ‘Selective whole-genome amplification reveals population genetics of Leishmania braziliensis directly from patient skin biopsies’, Pilling et al for the first time apply selective whole genome amplification (SWGA) on skin biopsies from Brazilian patients infected with Leishmania braziliensis. The manuscript is largely a proof-of-principle study that SWGA can be also applied to Leishmania, similar to its application to many other pathogens. The first three figures of this manuscript are dedicated to critically assess this method and show that SWGA (i) allows for over 60-fold enrichment of parasite over host DNA in synthetic, ‘spiked’ samples, with the same primer set working across all major parasite species, (ii) allows to enrich parasite DNA from samples obtained from infected mice and patients attaining a 10x coverage for 80% of the genome, and (iii) allow to monitor somy variation in spiked samples while being less resolutive on clinical material. They then apply the method on archived samples, a series of newly sequenced genomes and available genomes published by others to (i) gain insight into structural genomic changes (SNP, ndels, frameshifts), (ii) carry out a population genomic analysis using a phylogenetic approach, and (iii) reveal unique, regional variants for L. braziliensis in Brazil. The paper is well written, and the results are well controlled and clearly presented, making a convincing case that SWGA will be useful for the Leishmania community. Enthusiasm is somewhat tempered by the limitation of this method caused by inherent, technical noise that sees to preclude genome analysis in a more quantitative way, such as changes in SNP frequency, in chromosome and gene copy number, which represents key information for GWAS and biomarker discovery. Nonetheless, the method is a powerful tool for the assessment of unique structural features of a given genome and to use this information to assess the evolutionary relationship between genomes.

Major comments:

Discussion: While the advantages of SWGA over the more cost-intensive, commercially available solutions are convincingly presented throughout the manuscript, a more critical discussion of the limitations of SWGA compared to Sure Select arrays should be added. This would also allow to better highlight the strengths and applications of SWGA. How are the methods comparing with respect to technical background and quantitative genome assessment on for example gene and chromosome CNVs, or variations in SNP frequencies?

Figure 2: Panel 2B should be either discussed at the beginning (it is discussed at the very end of this chapter) or the panel labelling and organization should be adapted and come at last. The data shown in panel 2D should be more critically assessed and the cause (technical) and consequences (no quantitative genetic information possible) of the important coverage fluctuations seen across the entire chromosomes outside the ‘complicated’ areas should be discussed.

Figure 3: There is a misconception here: a SNP frequency ratio of 0.5 is not proof of a disomic state! Here, populations are sequenced and thus the somie values of each individual parasite are integrated in the final genome sequence. Thus, if the trisomy is caused by either of the two haplotypes, the heterozygous SNPs will be balanced again at 0.5 in the population. Also, for a meaningful analysis of SNP frequencies, a robust assessment of the read depth needs to be available, which is not the case for SWGA. I would remove this figure or add it to supp data, indicating that SWGA is less useful in estimating somie differences based on either read depth variations or SNP frequency shifts.

Figure 4: How does the nested procedure – i.e. a first shallow sequencing followed by sequencing of pooled libraries – affect SNP frequencies and read depth (see comments above)? Shouldn’t this introduce a major bias given that the initially obtained 20% includes both unique (hence the pooling), but also overlapping regions that will then be over-represented in the second round of sequencing? How does this affect the more quantitative aspects of comparative genome analyses, which are at the core of biomarker discovery? These drawbacks need to be discussed.

Figure 5 panel E: Define the triangle and the black dot in the legend. Based on the color code, all SWGA samples are clustering together. This is not properly highlighted in the results section. This signal should be pointed out and a role (even if minor) of SWGA on clustering should be discussed.

Figure 6: The functional enrichment observed for the structural mutants that change the protein sequence changes is potentially very interesting! The fact that half of the unique variants are contributed by the SWGA samples is a bit concerning – how sure is it that these are not introduced by the method? It is unclear to me how many samples show mutations in a given gene (same position or different), which would indicate convergence and thus selection. Have any of these genes been previously linked to treatment failure? I am surprised that these changes were not correlated to the observed and documented outcome of treatment across the samples with available genomes (including the 59 genomes sequenced by others). Is there any correlation, even if statistically insignificant?

**Part III – Minor Issues: Editorial and Data Presentation Modifications**

Reviewer #1: 1) Figure 4 is unnecessary

2) I believe the argument for lack of L. braziliensis sequences (discussion section) is not lack of infrastructure or difficulty in isolation, but rather a limitation of technical capacity and the most important, lack of economic resources for expensive sequencing projects.

Reviewer #2: In the manuscript ‘Selective whole-genome amplification reveals population genetics of Leishmania braziliensis directly from patient skin biopsies’, Pilling et al for the first time apply selective whole genome amplification (SWGA) on skin biopsies from Brazilian patients infected with Leishmania braziliensis. The manuscript is largely a proof-of-principle study that SWGA can be also applied to Leishmania, similar to its application to many other pathogens. The first three figures of this manuscript are dedicated to critically assess this method and show that SWGA (i) allows for over 60-fold enrichment of parasite over host DNA in synthetic, ‘spiked’ samples, with the same primer set working across all major parasite species, (ii) allows to enrich parasite DNA from samples obtained from infected mice and patients attaining a 10x coverage for 80% of the genome, and (iii) allow to monitor somy variation in spiked samples while being less resolutive on clinical material. They then apply the method on archived samples, a series of newly sequenced genomes and available genomes published by others to (i) gain insight into structural genomic changes (SNP, ndels, frameshifts), (ii) carry out a population genomic analysis using a phylogenetic approach, and (iii) reveal unique, regional variants for L. braziliensis in Brazil. The paper is well written, and the results are well controlled and clearly presented, making a convincing case that SWGA will be useful for the Leishmania community. Enthusiasm is somewhat tempered by the limitation of this method caused by inherent, technical noise that sees to preclude genome analysis in a more quantitative way, such as changes in SNP frequency, in chromosome and gene copy number, which represents key information for GWAS and biomarker discovery. Nonetheless, the method is a powerful tool for the assessment of unique structural features of a given genome and to use this information to assess the evolutionary relationship between genomes.

Major comments:

Discussion: While the advantages of SWGA over the more cost-intensive, commercially available solutions are convincingly presented throughout the manuscript, a more critical discussion of the limitations of SWGA compared to Sure Select arrays should be added. This would also allow to better highlight the strengths and applications of SWGA. How are the methods comparing with respect to technical background and quantitative genome assessment on for example gene and chromosome CNVs, or variations in SNP frequencies?

Figure 2: Panel 2B should be either discussed at the beginning (it is discussed at the very end of this chapter) or the panel labelling and organization should be adapted and come at last. The data shown in panel 2D should be more critically assessed and the cause (technical) and consequences (no quantitative genetic information possible) of the important coverage fluctuations seen across the entire chromosomes outside the ‘complicated’ areas should be discussed.

Figure 3: There is a misconception here: a SNP frequency ratio of 0.5 is not proof of a disomic state! Here, populations are sequenced and thus the somie values of each individual parasite are integrated in the final genome sequence. Thus, if the trisomy is caused by either of the two haplotypes, the heterozygous SNPs will be balanced again at 0.5 in the population. Also, for a meaningful analysis of SNP frequencies, a robust assessment of the read depth needs to be available, which is not the case for SWGA. I would remove this figure or add it to supp data, indicating that SWGA is less useful in estimating somie differences based on either read depth variations or SNP frequency shifts.

Figure 4: How does the nested procedure – i.e. a first shallow sequencing followed by sequencing of pooled libraries – affect SNP frequencies and read depth (see comments above)? Shouldn’t this introduce a major bias given that the initially obtained 20% includes both unique (hence the pooling), but also overlapping regions that will then be over-represented in the second round of sequencing? How does this affect the more quantitative aspects of comparative genome analyses, which are at the core of biomarker discovery? These drawbacks need to be discussed.

Figure 5 panel E: Define the triangle and the black dot in the legend. Based on the color code, all SWGA samples are clustering together. This is not properly highlighted in the results section. This signal should be pointed out and a role (even if minor) of SWGA on clustering should be discussed.

Figure 6: The functional enrichment observed for the structural mutants that change the protein sequence changes is potentially very interesting! The fact that half of the unique variants are contributed by the SWGA samples is a bit concerning – how sure is it that these are not introduced by the method? It is unclear to me how many samples show mutations in a given gene (same position or different), which would indicate convergence and thus selection. Have any of these genes been previously linked to treatment failure? I am surprised that these changes were not correlated to the observed and documented outcome of treatment across the samples with available genomes (including the 59 genomes sequenced by others). Is there any correlation, even if statistically insignificant?

PLOS authors have the option to publish the peer review history of their article (what does this mean?). If published, this will include your full peer review and any attached files.

Reviewer #1: **Yes: **Maria Adelaida Gómez

Reviewer #2: No
---

## [Decision Letter · Decision Letter 1]

22 Feb 2023

Dear Dr. Beiting,

We are pleased to inform you that your manuscript 'Selective whole-genome amplification reveals population genetics of Leishmania braziliensis directly from patient skin biopsies' has been provisionally accepted for publication in PLOS Pathogens.

Best regards,

Dawn Marie Wetzel

Guest Editor

PLOS Pathogens

Margaret Phillips

Section Editor

PLOS Pathogens

Kasturi Haldar

Editor-in-Chief

PLOS Pathogens

orcid.org/0000-0001-5065-158X

Michael Malim

Editor-in-Chief

PLOS Pathogens

orcid.org/0000-0002-7699-2064

Reviewer Comments (if any, and for reference):

Reviewer's Responses to Questions

**Part I - Summary**

Reviewer #1: The authors have adequately addressed all comments raised. The paper has much improved.

Reviewer #2: The authors responded to all my queries.

**Part II – Major Issues: Key Experiments Required for Acceptance**

Reviewer #1: (No Response)

Reviewer #2: The authors responded to all my queries.

**Part III – Minor Issues: Editorial and Data Presentation Modifications**

Reviewer #1: (No Response)

Reviewer #2: The authors responded to all my queries.

PLOS authors have the option to publish the peer review history of their article (what does this mean?). If published, this will include your full peer review and any attached files.

Reviewer #1: **Yes: **Maria Adelaida Gómez

Reviewer #2: No

---

## [Editor Report · Acceptance letter]

16 Mar 2023

Dear Dr. Beiting,

We are delighted to inform you that your manuscript, "Selective whole-genome amplification reveals population genetics of Leishmania braziliensis directly from patient skin biopsies," has been formally accepted for publication in PLOS Pathogens.

Best regards,

Kasturi Haldar

Editor-in-Chief

PLOS Pathogens

orcid.org/0000-0001-5065-158X

Michael Malim

Editor-in-Chief

PLOS Pathogens

orcid.org/0000-0002-7699-2064